# The biosynthetic pathway to tetromadurin (SF2487/A80577), a polyether tetronate antibiotic

**Rory F. Little** *¤, **Markiyan Samborskyy, Peter F. Leadlay**

Department of Biochemistry, University of Cambridge, Cambridge, United Kingdom

¤ Current address: Leibniz Institute of Natural Products Research and Infection Biology, Jena, Germany
* rory.little@leibniz-hki.de

**Data Availability Statement:** All relevant data are within the manuscript and its Supporting Information files. The genome sequence of A. verrucosospora has been deposited on GenBank (Accession number: CP053892).

## Abstract

The type I polyketide SF2487/A80577 (herein referred to as tetromadurin) is a polyether tetronate ionophore antibiotic produced by the terrestrial Gram-positive bacterium *Actinomadura verrucosospora*. Tetromadurin is closely related to the polyether tetronates tetronasin (M139603) and tetronomycin, all of which are characterised by containing a tetronate, cyclohexane, tetrahydropyran, and at least one tetrahydrofuran ring. We have sequenced the genome of *Actinomadura verrucosospora* to identify the biosynthetic gene cluster responsible for tetromadurin biosynthesis (the *mad* gene cluster). Based on bioinformatic analysis of the 32 genes present within the cluster a plausible biosynthetic pathway for tetromadurin biosynthesis is proposed. Functional confirmation of the *mad* gene cluster is obtained by performing in-frame deletions in each of the genes *mad10* and *mad31*, which encode putative cyclase enzymes responsible for cyclohexane and tetrahydropyran formation, respectively. Furthermore, the *A. verrucosospora* Δmad10 mutant produces a novel tetromadurin metabolite that according to mass spectrometry analysis is analogous to the recently characterised partially cyclised tetronasin intermediate lacking its cyclohexane and tetrahydropyran rings. Our results therefore elucidate the biosynthetic machinery of tetromadurin biosynthesis and lend support for a conserved mechanism of cyclohexane and tetrahydropyran biosynthesis across polyether tetronates.

## Introduction

SF2487/A80577 (**1**) (referred to in this manuscript as tetromadurin) is a type I polyketide polyether tetronate isolated independently from *Actinomadura verrucosospora* by two industrial research groups in the early 1990s [1, 2] (**Fig 1**). It belongs to the same family of polyether tetronates as tetronasin (**2**) (produced by *Streptomyces longisporoflavus*) and tetronomycin (**3**) (produced by *Streptomyces* sp. NRRL 11266), which are all characterised by containing a tetronate, cyclohexane, tetrahydropyran, and at least one tetrahydrofuran ring (**Fig 1**). Tetronasin and tetronomycin are notable for being near- "mirror image" versions of one another, containing the opposite configuration at each of ten equivalent stereocenters. Tetromadurin contains

**Funding:** The author(s) received no specific funding for this work.

**Competing interests:** The authors have declared that no competing interests exist.

structural elements of each, its cyclohexane and tetrahydropyran rings of tetromadurin having identical configuration to those in tetronasin, while its tetronate ring contains a tetronomycin-like exocyclic double bond. All three compounds, like other polyethers such as monensin, are ionophore antibiotics that disrupt ion signalling across cell membranes [1, 3–5]. In addition, tetromadurin has also demonstrated antiretroviral and antimalarial activity [1, 6].

The high degree of structural similarity between tetromadurin and tetronasin/tetronomycin suggests a similar biosynthetic pathway for its biosynthesis. The biosynthetic gene clusters (BGCs) responsible for the production of tetronasin (the *tsn* gene cluster) and tetronomycin (the *tmn* gene cluster) have both been sequenced (Genbank: FJ462704 and AB193609, respectively) [7, 8]. The *tsn* and *tmn* gene clusters both encode five type I polyketide synthase (PKS) enzymes that condense six (*2S*)-methylmalonyl-CoA and seven malonyl-CoA extension units to form the 26 carbon skeletons of tetronasin and tetronomycin. In addition, both gene

**Fig 1. The polyether tetronates tetromadurin (1), tetronasin (2), and tetronomycin (3).**

clusters also encode enzymes for creation of the four ring types, an *O*-methyltransferase for the C26 methoxy group, and a cyctochrome P450 for the primary hydroxyl group (C30 in tetronasin **2** and C28 in tetronomycin **3**) [7, 8]. In contrast, on the basis of its structure we predicted that the 31-carbon polyketide skeleton of tetromadurin is synthesised using six malonyl-CoA, eight (*2S*)-methylmalonyl-CoA, and one uncommon (*2R*)-methoxymalonyl-ACP (acyl-carrier protein) unit (**S1 Fig**). Also, unlike tetronasin and tetronomycin, tetromadurin contains two primary hydroxyl groups (positions C36 and C38) and two tetrahydrofuran rings.

We have performed whole-genome sequencing of *A. verrucosospora* to identify the biosynthetic gene cluster responsible for tetromadurin biosynthesis (the *mad* gene cluster). Bioinformatic analysis of the individual genes within the *mad* gene cluster enabled us to propose a complete pathway for tetromadurin biosynthesis. Furthermore, the PKS enzymes encoded by the *mad* gene cluster indicate an unusual case where one of the ketoreductase (KR) domains appears to act externally from its parent module. To functionally validate the *mad* gene cluster, independent in-frame deletions were made in the putative cyclase genes *mad10* and *mad31*. Deletion of either of these genes abolished tetromadurin production. In addition, mass spectrometry analysis of the *A. verrucosospora* Δmad10 mutant indicated the production of a late-stage tetromadurin biosynthetic intermediate lacking the cyclohexane and tetrahydropyran rings and one primary hydroxyl group, analogous to the intermediate recently characterised from a *S. longisporoflavus* Δtsn11 mutant [7]. Taken together, our findings clarify several aspects of tetromadurin biosynthesis, including identifying its biosynthetic gene cluster (BGC), determining the relative order of the two cytochrome P450-catalyzed hydroxylations, and supporting a conserved mechanism of cyclohexane and tetrahydropyran formation in polyether tetronate biosynthesis.

## Methods

### Oligonucleotides, plasmids, and bacterial strains

All oligonucleotide primers were synthesised by Sigma-Aldrich and are presented in **Table 1**. The plasmids and bacteria strains used/created in this study are presented in **Tables 2** and **3**, respectively.

### Growth and maintenance of actinomycete cultures

All actinomycete mycelial liquid cultures were grown at 30˚C at 200 rpm in conical flasks. The liquid medium was filled to no more than 1/5 of the total flask volume. A steel spring was placed at the bottom of each flask that extended around the inside face, preventing the culture from clumping. Foam bungs were used to seal the flask. *Actinomadura verrucosospora* was maintained on oatmeal agar (20 g/L oatmeal, 20 g/L agar) while *S. longisporoflavus* was maintained on TWM medium (5 g/L D-glucose, 10 g/L sucrose, 5 g/L tryptone, 2.5 g/L yeast extract, 0.036 g/L EDTA, 15 g/L agar, pH 7.1.). *S.* sp NRRL 11266 was maintained on SFM medium (20 g/L soy flour, 20 g/L D-mannitol, 20 g/L agar).

### Genomic DNA extraction

For the preparation of genomic DNA (gDNA), 250–500 μL of mycelia was centrifuged in a 1.5 mL microcentrifuge tube to pellet the cells and the supernatant was discarded. The cell pellet was resuspended in 500 μL of SET buffer (20 mM TrisCl, 75 mM NaCl, 75 mM EDTA, pH 7.2) containing 10 μL of lysozyme solution (50 mg/mL). After incubation at 37˚C for 1 h, 60 μL of 10% SDS (w/v) and 10 μL of proteinase K (20 mg/mL) were added and the tube was incubated at 55˚C for an additional 2 h. The sample was then mixed with 300 μL of 5 M NaCl

**Table 1. Primers used in this study.**

| Primer | Sequence (5'-3') |
|---|---|
| *Primers used for creating pYH7 gene deletion constructs* | |
| mad10_Up_Fw | GGGACTGATCAAGGCGAATACTTCAGCCGAGCTGCTGCTGGAACTGCTGC |
| mad10_Up_Rv | GCGGCGCGCGACGCGGCGCCGACGCGGTCATGC |
| mad10_Dn_Fw | GCGTCGGCGCCGCGTCGCGCGCCGCAGCC |
| mad10_Dn_Rv | GGGACCCGCGCGGTCGATCCCCGCACTTGATCAGGCCGGTGATGCCGGCC |
| mad31_Up_Fw | GGGACTGATCAAGGCGAATACTTCACCGGCAGCGGCGGAAGC |
| mad31_Up_Rv | CGACGATGTAGATCGGCGACACGCCGG |
| mad31_Dn_Fw | GGCGTGTCGCCGATCTACATCGTCGGAACGGG |
| mad31_Dn_Rv | GGGACCCGCGCGGTCGATCCCCGCACACCGGTGCCTCAGCG |
| tmn8_Up_Fw | GGGACTGATCAAGGCGAATACTTCAGCCGAGCTGCTGCTGGAACTGCTGC |
| tmn8_Up_Rv | GCGGCGCGCGACGCGGCGCCGACGCGGTCATGC |
| tmn8_Dn_Fw | GCGTCGGCGCCGCGTCGCGCGCCGCAGCC |
| tmn8_Up_Rv | GGGACCCGCGCGGTCGATCCCCGCACTTGATCAGGCCGGTGATGCCGGCC |
| *Primers used for screening A. verrucosospora exconjugant gDNA for targeted in-frame deletions* | |
| mad10_KO_Fw | CCTTCAAGAGATCCCCGAGG |
| mad10_KO_Rv | CGTCATCGGAACTCCCTGG |
| mad31_KO_Fw | CTCCTTCTGTTTGGGTGACG |
| mad31_KO_Rv | GAGTGCTTCAGGTGAGAACG |
| tmn8_KO_Fw | GGTCTCAGCCCGTCGTTC |
| tmn8_KO_Rv | TGTGCTGCTCGACCTGTC |
| *Primers used for creating pIB139 genetic complementation constructs* | |
| pIB139_mad10_Fw | AGGATCCACATATGTTGGGGATCCTATGAGCGATTCGGTTG |
| pIB139_mad10_Rv | GATATCGCGCGCGGCCGCGGATCCTTCACGCCGCAGGTTCC |
| pIB139_mad31_Fw | CCACATATGTTGGGGATCCTATGGGCATGACCTCGC |
| pIB139_mad31_Rv | CGCGCGCGGCCGCGGATCCTTCATCCGAAGGAAATGG |

and 500 μL of chloroform and centrifuged at 2200 x *g* for 15 min. The aqueous upper layer was transferred to a fresh microcentrifuge tube using a T1000 pipette tip with the end cut off to avoid shearing the gDNA. To precipitate the gDNA, 0.6x volumes of isopropanol was added followed by gentle mixing. The precipitated gDNA was washed twice in 70% ethanol before being air dried for 5–10 min. The gDNA pellet was dissolved in 100–200 μL of distilled water.

**Table 2. Plasmids used in this study.**

| Name | Description | Source |
|---|---|---|
| pYH7 | *tsr*, *bla*, *aac(3)IV*, *cos*, *oriT*, PT7, PT3, *oripIJ101*, *oriColEI*. In frame deletion in *Streptomyces* by homologous recombination | [8] |
| pIB139 | aac(3)IV, oriT, attP (ΦC31), int, PermE*, oripUC. Integrative plasmid for *in trans* complementation | [10] |
| pYH7-*tmn8* | Gene deletion construct for *tmn8* | This study |
| pYH7-*mad10* | Gene deletion construct for *mad10* | This study |
| pYH7-*mad31* | Gene deletion construct for *mad31* | This study |
| pIB139-*mad10* | Construct for the heterologous expression of *mad10* | This study |
| pIB139-*mad31* | Construct for the heterologous expression of *mad31* | This study |

**Table 3. Bacteria strains used in this study.**

| Name | Description | Source |
|---|---|---|
| *Actinomadura verrucosospora* NRRL B-18236 | Tetromadurin (SF2487/A80533) (**1**) producer | NRRL |
| *Streptomyces longisporoflavus* 83E6 | Tetronasin (**2**) producer | [7] |
| *Streptomyces* sp. NRRL 11266 | Tetronomycin (**3**) producer | [8] |
| *Actinomadura verrucosospora* Δmad10 | *A. verrucosospora* in which 906/1404 bp of the *mad10* coding frame has been deleted. No longer produces tetromadurin. Producer of T-17. | This study |
| *Actinomadura verrucosospora* Δmad31 | *A. verrucosospora* in which 291/561 bp of the *mad31* coding frame has been deleted. No longer produces tetromadurin. | This study |
| *Streptomyces. longisporoflavus* Δtsn11 | *S. longisporoflavus* 83E6 in which 1356 nt of the *tsn11* coding frame has been deleted. Producer of **17** | [7] |
| *Streptomyces. longisporoflavus* Δtsn15 | *S. longisporoflavus* 83E6 in which the entire 621 bp *tsn15* coding frame has been deleted | [7] |
| *Streptomyces. longisporoflavus* Δtsn15-mad31 | *S. longisporoflavus* Δtsn15 containing the integrative pIB139-*mad31* plasmid. | This study |
| *Streptomyces. longisporoflavus* Δtsn15-tmn8 | *S. longisporoflavus* Δtsn15 containing the integrative pIB139-*tmn8* plasmid. | This study |
| *Streptomyces.* sp NRRL 11266 Δtmn8 | *S.* sp NRRL 11266 in which the entire 573 bp *tmn8* coding frame has been deleted. No longer produces tetronomycin. | This study |

## Genome sequencing

Nextera shotgun and Nextera mate-pair libraries were constructed from high molecular weight genomic DNA isolated from *Actinomadura verrucosospora* NRRL-B18236. Sequencing was carried out on an Illumina MiSeq platform using the Illumina V2 500 cycles kit in 2 × 250 bp mode. Reads were processed using a custom adapter trimming tool(fastq_miseq_trimmer). Read pairs were then preassembled using FLASH v1.2.11 (https://ccb.jhu.edu/software/FLASH). For *de novo* assembly we used newbler v3.0. Several assemblies were carried out using either all or subsets of the input dataset, and the best assembly was selected using a score calculated from scaffold N50, edge and total number of contigs. The best assembly was polished using Pilon. ORFs were predicted *ab initio* using a customised version of the FGENESB pipeline V2.0 (2008) [www.softberry.com] and BLAST-searched against a filtered NCBI NR and KEGG datasets. Customised linguistic analysis was used for transfer of gene annotations. Annotation results were saved in EMBL format and manually curated in Artemis (http://www.sanger.ac.uk/science/tools/artemis). The revised genome sequence has been deposited in GenBank under accession number CP053892. The tetromadurin biosynthesis gene cluster (*mad* gene cluster) can be found within the genome using the accession CP053892:2037887–2164555.

## Revised sequence of Mad10

Close examination of the *mad10* gene revealed an alternative ATG start codon 15 codons upstream (corresponding to the amino acids MSDSVVIIGAGPVGL) to the start codon presented in Genbank: QKG20158. This alternative start codon was preferred as including these 15 extra amino acids improved its alignment with homologues proteins (see **S12 Fig** in the supplementary material for details), suggesting it is the true start codon.

For clarity, the revised sequence of Mad10 used for alignments and expressed from pIB139-*mad10* is presented below (bolded residues highlight the additional amino acid residues):

**MSDSVVIIGAGPVGL**MLAHELALAGVRTVVIERRPEIDARTVSGLIHERSVELLEQRG
LMEQIRREDGEPLVWDRLHFASFWLDMSELAKTDHSVVLLQTRIQRLLSDRAAARGVH

```
ILRRHELVGLSQDEDGVTARVHSPLGEEEIRCGYLVGCDGEDSAVRELAGFAVTRSGPS
WYGLLADVGSYAGPVGAGSFHEGGMFGQFGDASTMFRLMTIEIGVEAPPAEQPVTLE
EVRASIERITGERPTVEEPLWLHRHGNVTILADEYRNGRVFLAGDAAHFQFHPAGHAV
TIGLQDAVNLGWKLAAKLQGWAPAGLLDSYDAERRPYGRRACVYGRAQMALLDTAD
GPSALREVFGELLDHDVVNRHLVRAATDARYPMGQAEDLVGRRVPLVSLSTPGGEVPV
AETLHAARGVLLDLSGGTEPPDVTGWKDRVDVVAAEPTPEIPASAVLVRPDGHIAWA
GEAVGEGLHTALAAWFGEPAA
```

## Creation of pYH7 and pIB139 constructs

The general method used to create a gene-deletion construct was to PCR amplify 2 kb regions upstream and downstream of the gene of interest. The DNA polymerase used was Phusion™ High Fidelity Master Mix with GC Buffer (New England Biolabs). The PCR primers used to amplify each 2 kb fragment contained regions of overlap with both *Nde*I-linearised pYH7 [8] and the other 2 kb fragment, enabling all three to be seamlessly joined using Hot Fusion DNA assembly [9]. The assembled DNA product was then transformed into chemically competent *E coli* NovaBlue cells. Colony PCR was used to identify *E. coli* clones containing the correctly assembled insert, which were then fully sequenced using Sanger sequencing. The same protocol was followed to create the pIB139 [10] constructs, except the vector was linearised using *Xba*I and only a single fragment was inserted.

## Intergeneric conjugation of *Actinomadura verrucosospora*, *Streptomyces longisporoflavus*, and *Streptomyces* sp. NRRL 11266

The DNA methylation deficient ET12567/pUZ8002 cells transformed with the pYH7 or pIB139 plasmid to be transferred into the actinomycete host were grown to an $A_{600}$ of 0.4–0.6 in 2TY (16 g/L tryptone, 10 g/L yeast extract, 5 g/L NaCl) medium containing selective antibiotics. The ET12567/pUZ8002 cells were then spun down and washed twice in 20 mL of 2TY medium before being resuspended in 300 μL of 2TY. The actinomycete strain to be conjugated (either as mycelia or spores) was prepared following the protocol from *Practical Streptomyces Genetics* [11]. After gentle mixing, the two bacteria types were plated onto 35 mL of SFM containing 10–20 mM $MgCl_2$ and left at 30˚C for 12–20 h. The surface of the plate was then flooded with 1 mL of MQ water containing 35 μL of apramycin (50 mg/mL) and 25 μL nalidixic acid (25 mg/mL). The plate was then incubated at 30˚C to promote the growth of exconjugants. Exconjugants were verified by restreaking onto SFM containing 30–50 μg/mL apramycin and 25 μg/mL nalidixic acid.

## Metabolite analysis

To detect the production of tetromadurin, *A. verrucosospora* was grown at 30˚C for seven days on oatmeal agar. To detect the production of tetronasin, *S. longisporoflavus* was grown at 30˚C on tsn medium B (30 g/L tryptic soy broth, 3 g/L $CaCO_4$, 100 g/L dextrin, 20 g/L agar, trace elements: 4 mg/L $FeSO_4$, 4 mg/L $ZnSO_4$, 0.6 mg/L $CuSO_4$, 0.4 mg/L $MnSO_4$, 0.4 mg/L $KMoO_4$). To detect the production of tetronomycin, *S.* sp NRRL 11266 was grown at 30˚C on SFM medium. Following the fermentation, the agar was cut into cubes and extracted by submerging in ethyl acetate. The ethyl acetate was then evaporated to dryness under reduced pressure and the organic extract was redissolved 1 mL of methanol and centrifuged at 20,000 g for 20 min to remove any particulates, followed by HPLC-MS analysis. For the analysis of small molecules, an HPLC (Hewlett Packard, Agilent Technologies 1200 series) coupled to a mass spectrometer (Thermo Finnigan MAT LTQ) was used. The HPLC-MS was fitted with a 250 mm x 4.6 mm 5μm C18 reverse-phase column (5μ OSD3, 100 Å. Phenomenex, USA). The

mobile phase consisted of 20 mM ammonium acetate and methanol. The mobile phase flow rate was 0.7 mL/min with the following gradient: 0–5 min, 5–75% methanol; 5–30 min, 75–95% methanol, 30–34 min, 95% methanol 35–36 min, 95–5% methanol. The mass spectrometer was operated in positive electrospray ionisation mode set to full scan (from m/z 200–2000).

## Construction of phylogenetic trees and protein sequence alignments

Maximum-likelihood (based on the Le-Cascuel model [12]) phylogenetic trees were created using MEGA7 [13] by following the protocol described in [14]. 1000 bootstrap replicates were performed. Positions with less than 95% site-coverage were excluded. Protein sequence alignments were performed using ClustalOmega [15].

## Results

### Identification of the mad gene cluster in *A. verrucosospora*

A strain of the *A. verrucosospora* NRRL B-18236 was obtained from the Agricultural Research Service Culture Collection (NRRL). Prior to sequencing, the competence of this strain as a tetromadurin producer was tested. *A. verrucosospora* NRRL B-18236 was fermented for seven days on oatmeal agar followed by LC-MS analysis of the organic extract, where tetromadurin ($[M+Na^+]$ = 783.8, $\lambda_{MAX}$ = 252 nm, 300 nm (MeOH)) was clearly detectable (**S2 Fig**).

Whole-genome sequencing was then performed using a combined shotgun and long-range mate pair MiSeq approach. The genome was assembled into a single 10.215 Mbp scaffold (Genbank: CP053892) and analysed by AntiSMASH 4.0 [16] to identify biosynthetic gene clusters. Only one of the detected biosynthetic gene clusters, ca. 110 kbp and featuring 32 genes (CP053892:2037887–2164555), possessed the biosynthetic features likely required for tetromadurin biosynthesis. The exact boundaries of the tetromadurin biosynthetic gene cluster (the *mad* gene cluster) were decided on the basis of flanking genes whose annotation suggested no obvious connection with tetromadurin biosynthesis (such as primary metabolism). The proposed functions of the genes in the *mad* gene cluster, predicted using BLAST [17], is presented in **Fig 2**. A diagram of the spatial organisation of the *mad* gene cluster is presented in **Fig 3**. The majority of genes within the *mad* gene cluster could be ascribed plausible roles in tetromadurin biosynthesis, enabling us to propose a complete biosynthetic pathway (**Figs 4** and **5**). The rationale for this biosynthetic pathway is discussed in the following sections.

### Polyketide synthase genes of the *mad* gene cluster

The *mad* gene cluster contains seven genes encoding type I polyketide synthase multienzymes (*madAI-madAVII*), collectively consisting of one loading and 14 extension modules. No thioesterase (TE) domain is present in any of the modules, as expected for a polyketide tetronate pathway [18]. The loading PKS module (MadAI_KS_LM) could be differentiated from the extension modules by its KS domain catalytic cysteine being replaced by a glutamine ($KS^Q$ domain) (**S3 Fig**) [19]. The order in which the remaining PKS enzymes process the tetromadurin intermediate was then predicted based on the agreement of the proposed product with the actual structure of tetromadurin. The linear polyketide predicted by ordering the PKS enzymes MadAI-MadAVII (**13**) (i.e, MadAI initiates tetromadurin biosynthesis and performs two extension cycles before transferring the nascent polyketide chain to downstream to MadAII etc. . .) is in near perfect agreement to the linear polyketide we predicted from retrobiosyntheic analysis of tetromadurin itself (**14**) (**Fig 6**). The placement of the C5-C6, C10-C11, C12 and C13 double bonds in **14** was guided by the recently isolated partially cyclised tetronasin intermediate [7].

| orf | Size* | Homologue and origin | Identity/ similarity (%) | Proposed function of encoded protein |
|---|---|---|---|---|
| mad1 | 130 | WP_067469187, *Actinomadura macra* | 79/90 | SARP activator protein |
| mad2 | 937 | KX263301 (MonH) *Streptomyces cinnamonensis* strain ST021 | 50/59 | LuxR transcriptional activator |
| mad3 | 192 | WP_017621714 (EntD), *Nocardiopsis gilve* | 66/73 | 4'-phosphopantetheinyl transferase |
| mad4 | 285 | WP_113699285 (CpdA), *Nonomuraea* sp. NEAU-YG30 | 74/80 | Metallophosphoesterase |
| madAI | 4869 | AJE80656, *Streptomyces albus* | 54/65 | Type I PKS: KS$^\circ$, ATp, ACP, KS, ATa, DH, KR, ACP, KS, ATa, DH, ER, ACP |
| mad6 | 256 | WP_110701439 (GrsT) *Streptosporangium* sp. caverna | 59/72 | Type II thioesterase |
| mad7 | 642 | WP_067456393, *Actinomadura macra* | 66/73 | FkbH-like glyceryl-S-ACP synthase |
| mad8 | 75 | WP_031047730, *Amycolatopsis albispora* | 63/90 | Acyl-carrier protein |
| madAVII | 1588 | WP_093461893, *Streptomyces melanosporofaciens* | 50/60 | Type I PKS: KS, ATa, KR, ACP |
| mad10 | 467 | ACR50781 (Tsn11), *Streptomyces longisporoflavus* | 48/60 | FAD-dependent monooxygenase/ putative [4+2] cyclase |
| mad11 | 288 | WP_037917628 (FabB), *Streptomyces* sp. PCS3-D2 | 69/75 | Glyceryl-S-ACP dehydrogenase |
| mad12 | 78 | PPS67085, *Streptomyces* sp. 46 | 55/69 | Acyl-carrier protein |
| mad13 | 362 | WP_027945620 (CaiA), *Amycolatopsis taiwanensis* | 71/78 | 2-hydroxy-3-oxopropionyl-S-ACP dehydrogenase |
| mad14 | 375 | WP_055537123, *Streptomyces neyagawaensis* | 71/80 | FkbH-like glyceryl-S-ACP synthase |
| mad15 | 226 | WP_086736314, *Streptomyces glaucescens* | 64/78 | SAM-dependent methyltransferase |
| mad16 | 342 | WP_113696535, *Amycolatopsis albispora* | 64/76 | FabH 3-oxoacyl-ACP synthase III family protein |
| mad17 | 281 | WP_055549000 (AceF), *Streptomyces kanamyceticus* | 58/75 | Acyltransferase |
| mad18 | 383 | WP_067456402, *Actinomadura macra* | 54/70 | Dehydratase |
| madC | 472 | CCD31908, *Streptomyces albus* | 55/70 | Epoxidase |
| mad20 | 273 | WP_119927171 *Streptosporangiaceae bacterium* YIM 75507 | 44/61 | Hypothetical protein |
| madAVI | 5623 | WP_049717538, *Streptomyces caatingaensis* | 50/62 | Type I PKS: KS, ATp, DH, ER, KR, ACP, KS, ATp, DH, KR, ACP, KS, ATm, DH, KR, ACP |
| madAV | 5670 | WP_045867303, *Streptomyces* sp. NBRC 110027 | 51/63 | Type I PKS: KS, ATp, DH, ER, KR, ACP, KS, ATp, DH, KR, ACP, KS, ATp, DH, KR, ACP |
| madAIV | 3343 | WP_104651095, *Streptomyces cinnamoneus* | 54/66 | Type I PKS: KS, ATp, DH, KR, ACP, KS, ATp, KR, ACP |
| madAIII | 1819 | WP_078876216, *Streptomyces* sp. 769 | 55/67 | Type I PKS: KS, ATa, DH, KR, ACP |
| mad25 | 535 | WP_092886900, *Actinopolymorpha cephalotaxi* | 57/72 | ABC transporter permease |
| mad26 | 312 | WP_026416314, *Actinomadura oligospora* | 69/81 | ABC transporter ATP-binding protein |
| mad27 | 180 | WP_092886896, *Actinopolymorpha cephalotaxi* | 52/69 | MarR family transcriptional regulator |
| madB | 132 | ACR50776 (TsnB), *Streptomyces longisporoflavus* | 53/69 | Epoxide hydrolase |
| mad29 | 386 | ACR50783 (Tsn12), *Streptomyces longisporoflavus* | 52/65 | Cytochrome P450 |
| mad30 | 400 | WP_055419038, *Streptomyces pactum* | 59/72 | Cytochrome P450 |
| mad31 | 186 | BAF73716 (Tmn8), *Streptomyces* sp. NRRL 11266 | 29/45 | Tetrahydrofuran-forming pericyclase |
| madAII | 3978 | WP_106436315, *Streptomyces bingchenggensis* | 57/68 | Type I PKS: KS, ATp, DH, KR, ACP, KS, ATp, DH, ER, ACP |

*Amino acid residues of encoded protein

**Fig 2. Predicted functions of genes in the *mad* gene cluster.** ATa = malonyl-CoA selective AT domain (acetate unit). ATp = (*2S*)-methylmalonyl-CoA selective AT domain (propionate unit). ATm = predicted (*2R*)-methoxymalonyl-ACP selective AT domain.

The substrate selectivity of the acyltransferase (AT) domains were predicted from the presence of specific amino acid motifs, particularly the presence of a (H/T/V/Y)AFH (acetate-incorporating), or (Y/V/W)ASH (propionate-incorporating) motif (**S4 Fig**) [20–22]. The AT domain of the loading module contains the hybrid HASH motif, previously found in modules with a relaxed selectivity that can incorporate both malonyl-CoA and (2*S*)-methylmalonyl-CoA [23]. However, no congener of tetromadurin with one fewer methyl group has been reported, indicating that (2*S*)-methylmalonyl-CoA is still specifically recruited by the loading module. The AT domain of module 13 (MadAVI_AT_13) contains a YASH motif, indicating selectivity for a (2*S*)-methylmalonyl-CoA. However, based on the structure of tetromadurin this module should actually incorporate (2*R*)-methoxymalonyl-ACP to form the C4 methoxy group. The selectivity motifs of AT domains that incorporate (2*R*)-methoxymalonyl-ACP are

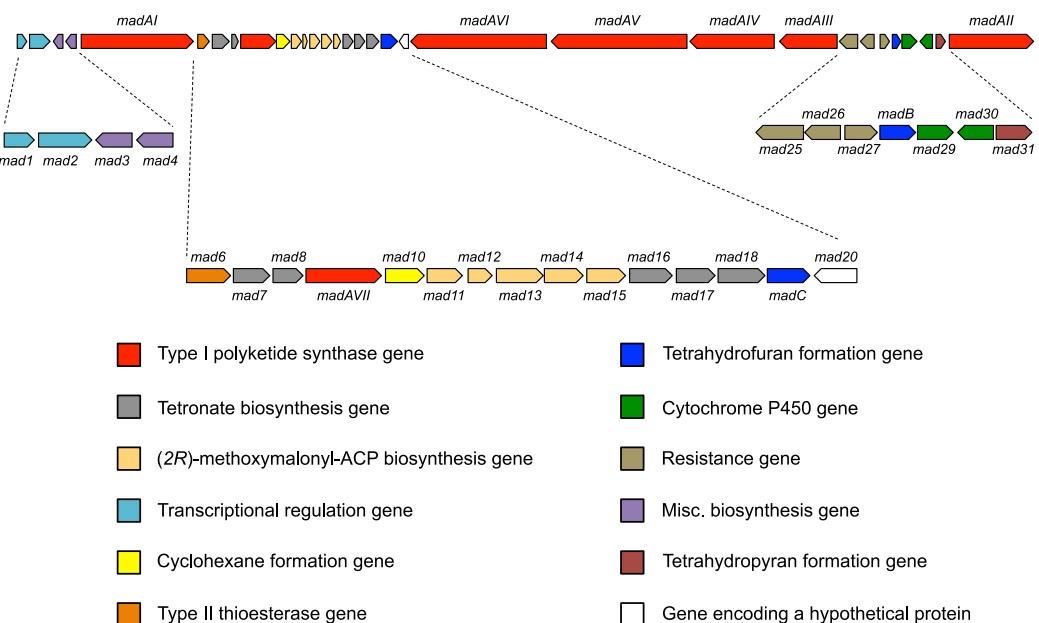

**Fig 3. The tetromadurin biosynthetic gene cluster (the _mad_ gene cluster).** The total size of the _mad_ gene cluster is 110 kbp. A total of 32 were predicted to comprise the _mad gene_ cluster. Genes are colour coded according to their predicted function.

poorly defined and typically resemble either a malonyl-CoA or (2_S_)-methylmalonyl-CoA selective domain [24]. However, given the _mad_ gene cluster contains the genes to synthesise (2_R_)-methoxymalonyl-ACP (discussed below), it is likely that MadAVI_AT_13 really is selective for this uncommon extension unit.

With one exception, the tally of reducing domains present in each module is also consistent with the structure of tetromadurin. Modules 2, 4, 8, and 11 all contain a ketoreductase (KR), dehydratase (DH), and enoylreductase (ER) domain, consistent with complete saturation at C27, C23, C15, and C9, respectively (**S5 Fig**). Since modules 4 and 11 incorporate propionate units, the ER domains in these modules also determine the configuration of the C33 and C37 α-methyl groups, respectively. These ER domains both contain amino acid motifs associated with the D-configuration (lacking the key tyrosine residue) [25], consistent with the D-configured α-methyl substituents at these positions (**S5 Fig**).

The DH domains of modules 1, 3, 6, 9, 10, and 13 are all appropriately placed to form the α-β double bonds at C29, C25, C19, C13, C11, and C5, respectively. Although module 5 contains a DH domain, C21 contains a hydroxyl group in the final tetromadurin structure, suggesting that MadAIII_DH_5 is inactive. Closer examination of MadAIII_DH_5 confirmed this inactivity, as it is missing the tyrosine from the YGP motif. The tyrosine side chain of the YGP motif in DH domains is proposed to assist binding to the β-hydroxyl group of the substrate [19] and has been shown experimentally to be essential [26]. The remaining DH domains all appear to be active (**S6 Fig**).

KR domains exert significant stereochemical influence over growing polyketide chains, determining the stereochemistry of the β-hydroxyl group and (where appropriate) the α-group of each extension unit. In the proposed MadAI-MadAVII module order, the KR domains of module 5, 7, and 12 are appropriately placed to form the hydroxyl groups at C21, C17, and C7, respectively. In the case of tetromadurin, all backbone β-hydroxyl groups and the α-methyl groups adjacent to these hydroxyls (C35 and C38) are in the D-configuration,

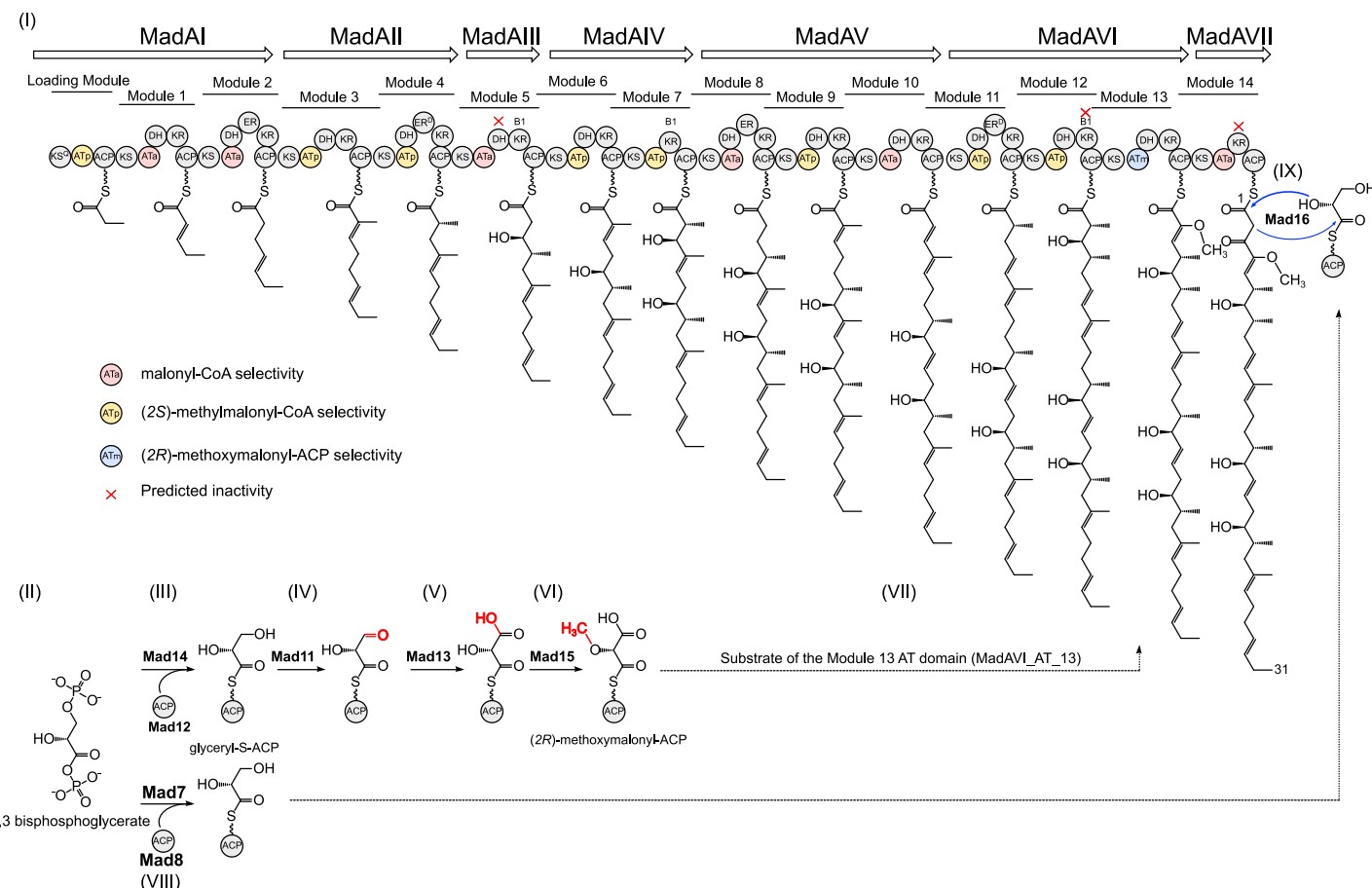

**Fig 4. The proposed biosynthesis pathway of tetromadurin (part 1).** (I) The seven PKS enzymes (MadAI-MadAVIII) and predicted linear tetromadurin intermediates of the *mad* gene cluster. ATa domains contain the amino acid motifs associated with malonyl-CoA (acetate unit) incorporation. ATp domains contain the amino acid motifs associated with (2S)-methylmalonyl-CoA (propionate unit) incorporation. The ATm domain is predicted to recognise (2R)-methoxymalonyl-ACP. ER^D domains are predicted to create a D-configured α-methyl group. B1 KRs are predicted to create D-configured α-methyl and β-hydroxyl groups. The KS domain in MadAI is a KS^Q domain. The red crosses indicate domains that are predicted to be inactive on the basis of their amino acid sequence. (II) The key biosynthetic precursor 1,3-bisphosphoglycerate, involved in both tetronate formation and (2R)-methoxymalonyl-ACP biosynthesis. (III) Formation of glyceryl-ACP by the FkbH-like enzyme Mad14 and the ACP Mad12. (IV) Oxidation of glyceryl-ACP by the dehydrogenase Mad11 to form 2-hydroxy-3-oxopropionyl-ACP. (V) Oxidation of 2-hydroxy-3-oxopropionyl-ACP by the dehydrogenase Mad13 to form hydroxymalonyl-ACP. (VI) Methylation of hydroxymalonyl-ACP by the *O*-methyltransferase Mad15 to form (2R)-methoxymalonyl-ACP. (VII) The (2R)-methoxylmalonyl-ACP is incorporated by the MadAVI_AT_13. (VIII) Synthesis of a second pool of glyceryl-ACP by FkbH-like Mad7 and ACP Mad8. (IX) Glyceryl-ACP is a substrate of the FabH-like protein Mad16 that catalyses tetronate formation and concomitant chain release from the PKS. Bonds/atoms have been coloured red to highlight the chemical change.

indicating the activity of B1 type KR domains [19]. The distinguishing features of a B1 KR domain are an (L,V,I)DD motif and the absence of a proline (a feature of B2-type KRs) two residues C-terminal of the catalytic tyrosine [27, 28]. Protein sequence analysis confirmed that the KR domains from modules 5 and 7 are indeed B1, but modules 12 and 14 and appear to be inactive (**S7 Fig**). The module 12 and 14 KR domains both lack the key catalytic tyrosine residue, with module 12 also containing a deletion in its NADPH binding site [19, 29]. While the inactivity of KR_14 is consistent with the C3 keto group of tetromadurin, the inactivity of KR_12 is not. An inactive KR domain at this position should result in a keto group at C7 rather than the C7 D-hydroxyl group observed in the tetromadurin (**Fig 6**). Module 12 also contains a seemingly active DH domain that should reduce a C7 D-hydroxyl by a C6-C7 *trans* double bond. How the tetromadurin C7 hydroxyl group is formed is therefore unclear.

## Genes for tetronate biosynthesis

The *mad* gene cluster contains homologues of all five glycerate-utilisation operon [18] genes: *mad7*, *mad8*, *mad16*, *mad17*, and *mad18*. *mad7* and *mad8* respectively encode a FkbH-like

**Fig 5. The proposed biosynthetic pathway of tetromadurin (part 2).** (X) In the first step of tetrahydrofuran formation, MadC catalyses epoxidation of the C24-C25 and C28-C29 *E* double bonds. (XI) The epoxide hydrolase MadB catalyses formation of the two tetrahydrofuran rings through a cascade epoxide ring-opening; (XII) Acetylation of the C41 hydroxyl group is catalysed by Mad17. (XIII) Elimination of the acetyl group by Mad18 forms the C40-C41 exocyclic double bond. (XIV) Hydroxylation of C36 by Mad29. It is possible this hydroxylation event takes place earlier in the biosynthesis (XV) Mad10 catalyses the formation of an oxadecalin containing intermediate **10**. (XVI) The equivalent oxadecalin intermediate in tetronasin biosynthesis acquired a water at C3 to form a hemiacetal [7]. Given the structural similarity between tetromadurin and tetronasin, it is likely the tetromadurin oxadecalin intermediate can be hydrated at C3 to form **11**. (XVII) Mad31 catalyses formation of the tetrahydropyran ring and dismantles the oxadecalin ring to form **12**. (XVIII) Mad30 then catalyses hydroxylation of C38, forming tetromadurin **1**. Bonds/atoms have been coloured red to highlight the chemical change.

Linear intermediate product predicted from MadAI-MadAVII (**13**):

Linear intermediate predicted from the tetromadurin structure (**14**):

**Fig 6. Hypothetical ACP-bound linear tetromadurin backbone. 13**: the hypothetical linear tetromadurin intermediate predicted to be produced by the MadAI-MadAVII arrangement. **14**: the hypothetical linear tetromadurin intermediate predicted from the structure of tetromadurin itself.

protein and a standalone acyl carrier protein (ACP) found in the biosynthetic pathways of other tetronates [8, 30]. Mad7 likely catalyses formation of glyceryl-ACP from 1,3-bisphosphoglycerate using Mad8 as the ACP scaffold. Mad16 is the FabH-like protein known for catalysing tetronate formation and chain release [18]. Finally, Mad17 and Mad18 are homologues of the acyltransferase Agg4 and the dehydratase Agg5, respectively, responsible for exocyclic double bond formation in agglomerin biosynthesis [31]. Mad17 likely catalyses the acetylation of the C41 hydroxyl, followed by Mad17 catalysing elimination of the acyl group to form the exocyclic C40-C41 double bond (**S8 Fig**).

## Genes for (2*R*)-methoxymalonyl-ACP biosynthesis

Several polyketide natural products are known to incorporate the unusual methoxymalonate extender unit derived from (2*R*)-methoxymalonyl-ACP [32], the first example being the macrocyclic immunosuppressant FK520 [24]. Five genes in the FK520 BGC were identified as responsible for the biosynthesis of (2*R*)-methoxymalonyl-ACP from 1,3-bisphosphoglycerate [24]. Homologues of these five genes have since been found in the BGCs of other natural products known to incorporate (2*R*)-methoxymalonyl-ACP, such as tautomycin [33], oxazolomycin [34], and geldanamycin [35, 36]. The *mad* gene cluster also contains homologues of these genes: *mad11*, *mad12*, *mad13*, *mad14*, and *mad15*, which appear to be part of a single operon. The gene *mad14* encodes a second FkbH-like protein, distinct from Mad8, that appears to catalyse formation of a second pool of glyceryl-ACP using 1,3 bisphosphoglycerate and Mad12, another standalone ACP. Whether the two pools of glyceryl-ACP can interchangeably be used in tetronate formation or (2*R*)-methoxymalonyl-ACP biosynthesis is unknown. The primary hydroxyl group of glyceryl-ACP undergoes two oxidation events to form (2*R*)-hydroxymalonyl-ACP. Based on its similarity to the dehydrogenase enzymes in the FK520 pathway, Mad11 is proposed to catalyse the first oxidation, forming 2-hydroxy-3-oxopropionyl-ACP. Mad13 then catalyses the second oxidation to form (2*R*)-hydroxymalonyl-ACP. The *O*-methyltransferase Mad15 likely converts (2*R*)-hydroxymalonyl-ACP into (2*R*)-methoxymalonyl-ACP (**S9 Fig**). The exact timing of *O*-methylation is uncertain, with some evidence suggesting it precedes the oxidation steps [37].

## Tetrahydrofuran ring formation

The *mad* gene cluster encodes an epoxidase (MadC) and an epoxide hydrolase (MadB), homologues of which are present in the BGCs of many other tetrahydrofuran-containing polyethers

[7, 8, 37–39]. In a mechanism first proposed for monensin [40], the epoxidase and epoxide hydrolase catalyse a regio- and stereospecific oxidation cyclisation to form one or multiple rings. It is likely that MadC catalyses the stereospecific epoxidation of the C24-C25 and C28-C29 *E* double bonds. MadB then catalyses opening of the two epoxide rings to form two tetrahydrofuran rings (**S10 Fig**). The timing of tetrahydrofuran formation in polyether is not certain, though evidence from other polyether pathways suggests it may occur whilst the intermediate is still bound to the PKS [7, 39, 41].

## Cyclohexane and tetrahydrofuran ring formation

The gene *mad10* is homologous to *tsn11* and *tmn9* from the *tsn* and *tmn* gene clusters, respectively. All three genes encode homologues of PyrE3, the [4+2] cyclase (Diels-Alderase) responsible for catalysing dialkyldecalin formation in the biosynthesis of pyrroindomycin A [42]. The discovery that PyrE3 catalyses an apparent [4+2] cycloaddition was surprising given its homology to MtmOIV-family FAD-dependent monooxygenase. Tsn11, despite also resembling an FAD-dependent monooxygenase, was recently demonstrated to catalyse an apparent inverse-electron demand hetero-Diels-Alder reaction to convert a tetronasin precursor **17** (in which the cyclohexane and tetrahydropyran rings have not yet formed) (**Fig 9C**) into an oxadecalin-containing intermediate [7]. We therefore propose that Mad10 catalyses an equivalent reaction in tetromadurin biosynthesis (**S11 Fig**). In support of this, Mad10, like Tsn11, also contains mutations in several of four conserved arginine residues involved in NADPH utilisation, indicating it is not a functional monooxygenase (**S12 Fig**) [7, 43, 44]. Further, inspection of a maximum-likelihood phylogenetic tree revealed that while Tmn9, Mad10, and Tsn11 form a clade together, they are more related to the PyrE3-like [4+2] cyclases [42, 42, 45–48] than to their homologues that are *bona fide* FAD-dependent monooxygenases [43, 49–51] (**S13 Fig**).

*mad31* is a homologue of *tsn15* and *tmn8*, and all are homologs of *vstJ*, which encodes an enzyme responsible for catalysing spirotetronate formation via an intramolecular [4+2] cyclo-addition in versipelostatin biosynthesis [46]. Tsn15 was recently demonstrated to catalyse the final step in tetronasin biosynthesis, dismantling the oxadecalin intermediate produced by Tsn11 and forming the tetrahydropyran ring [7]. The crystal structure of Tsn15 has been solved and a tryptophan residue essential for substrate binding identified [7]. Mad31 contains a tryptophan at the equivalent locus, suggesting that tetromadurin biosynthesis also proceeds via an oxadecalin intermediate [7] (**S14 Fig**).

## Hydroxylation of C36 and C38

Two cytochrome P450 enzymes are encoded in the *mad* gene cluster. One of these, Mad29, is homologous to Tsn12 (51% identity) and Tmn14 (32% identity)—the cytochrome P450s encoded by the *tsn* and *tmn* gene clusters, respectively [7, 8]. Tmn14 is predicted to catalyse the hydroxylation of C28 in tetronomycin while Tsn12 is predicted to catalyse the hydroxylation of the equivalent carbon (C30) in tetronasin biosynthesis (**S15 Fig**) [7, 8]. By analogy, Mad29 is therefore proposed to catalyse hydroxylation of C36 in tetromadurin, leaving the second cytochrome P450, Mad30, to catalyse hydroxylation of C38 (**S15 Fig**). No ferredoxin gene is present in the *mad* gene cluster, so presumably one encoded elsewhere in the genome is used to regenerate the cytochrome cofactor.

## Other genes in the *mad* gene cluster

Finally, the *mad* gene cluster contains several other genes likely involved in transcriptional regulation or export of tetromadurin itself. On one end of the *mad* gene cluster the genes *mad1* and *mad2* encode predicted SARP [52, 53] and LuxR [54] transcriptional regulators,

respectively. Another gene, *mad27*, encodes a putative MarR (multiple antibiotic resistance regulator)-like transcriptional regulator. First described in *E. coli*, MarR is a transcriptional regulator of several genes that confer antibiotic resistance [55]. Adjacent to *mad27* are the genes *mad25* and *mad26* that encode an ABC transporter and an ABC transporter ATP-binding protein, respectively, both of which are likely involved in exporting tetromadurin from the cytoplasm [56]. Mad27 may induce expression of *mad25* and *mad26* in the presence of tetromadurin, resulting in self-resistance [57, 58].

The gene *mad3* encodes a 4'-phosphopantetheinyl transferase, required for activating ACP domains by attaching a 4'-phosphopantetheine prosthetic group. Adjacent to *mad3* is the phosphoesterase *mad4*, homologues of which are found in BGCs of other polyketides [8, 59], the products of which may hydrolyse ACP-bound 4'-phosphopantetheine groups [59–61]. *mad6* is predicted to encode type II thioesterase, likely having an "editing" role in hydrolysing PKS active sites containing mis-acylated intermediates [62, 63]. The only gene that could not be assigned a plausible function is *mad20*, which encodes a hypothetical protein with no characterised homologues.

## Creation of in-frame deletion mutants in *mad10* and *mad31*

To verify experimentally that the *mad* gene cluster is indeed responsible for tetromadurin production (as the bioinformatics-based analysis strongly suggested), we performed in-frame deletions in the putative cyclase genes *mad10* and *mad31*. Each gene was individually knocked out in *A. verrucosospora* using a homologous recombination method, creating *A. verrucosospora* Δmad10 and *A. verrucosospora* Δmad31 respectively (S16 Fig). Tetromadurin production was completely abolished in the *A. verrucosospora* Δmad10 mutant, indicating that, like Tsn11 and its homologue encoded in the *tmn* gene cluster, Tmn9, Mad10 is an essential biosynthetic enzyme. To attempt to rescue tetromadurin production in *A. verrucosospora* Δmad10, *mad10* was introduced back into the strain *in trans* on the ΦC31 integrative plasmid pIB139 [10], resulting in tetromadurin production at 28% of wild type levels (Fig 7). The abolition of tetromadurin production in the *A. verrucosospora* Δmad10 and its subsequent rescue provides clear functional validation that the *mad* gene cluster is responsible for tetromadurin biosynthesis.

Tetromadurin production in the *A. verrucosospora* Δmad31 mutant was also significantly decreased, producing tetromadurin at just 3% of wild type levels (Fig 7), indicating that, like its homologue Tsn15, it is an essential biosynthetic enzyme [7]. However, *in trans* complementation of *mad31* using pIB139 did not rescue tetromadurin production (Fig 7). The failed *mad31* complementation, paired with the *mad10* complementation only restoring tetromadurin production in *A. verrucosospora* Δmad10 to 28% wild type levels, suggest expression issues of the streptomycete-optimised vector pIB139 within *A. verrucosospora*.

The finding that the *A. verrucosospora* Δmad10 no longer produces tetromadurin, as previously shown for the analogous *S.* sp NRRL Δtmn9 and *S. longisporoflavus* Δtsn11 mutants [7, 8], confirms that the PyrE3-like [4+2] cyclase family is essential for the biosynthesis of all three polyether tetronates. Likewise, *A. verrucosospora* Δmad31 and the previously analysed *S. longisporoflavus* Δtsn15 [7] mutant indicate that a conserved VstJ-like cyclase is also essential for polyether tetronate biosynthesis. In parallel, we also analysed the consequences of deleting *tmn8*, the *mad31/tsn15* homologue from the *tmn* gene cluster. *S.* sp NRRL 11266 Δtmn8 was created and analysed for tetronomycin production. Fermentation and analysis of the organic extract of this strain confirmed it no longer produces tetronomycin, confirming that VstJ-like cyclases are also essential for the biosynthesis of all three polyether tetronates (S17 and S18 Figs).

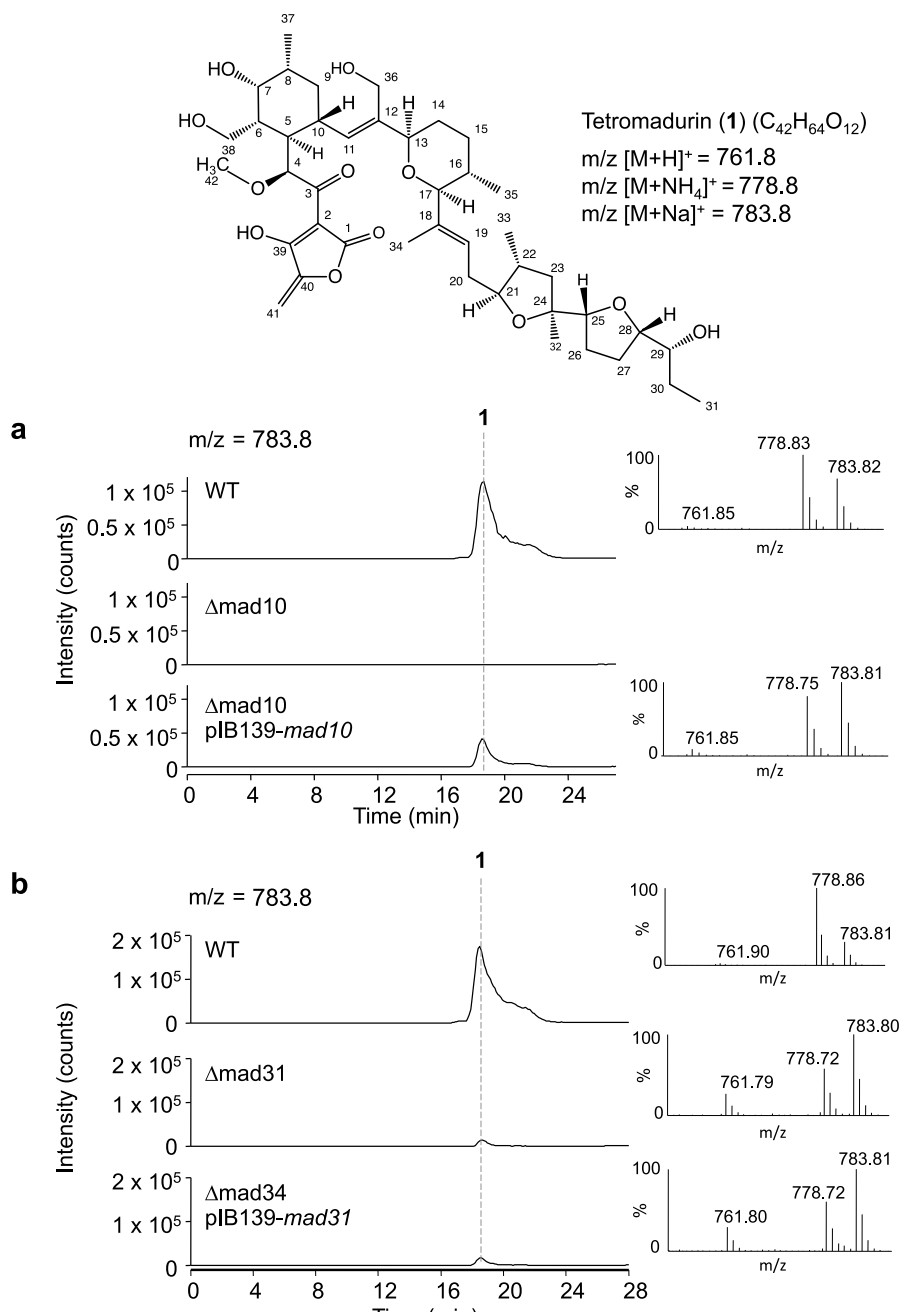

**Fig 7. HPLC-MS analysis of *A. verrucosospora* Δmad10 and *A. verrucosospora* Δmad31. a,** The structure and molecular weight of tetromadurin (**1**) produced by *A. verrucosospora*. **b,** Extracted m/z = 783.8 HPLC-MS spectra from *A. verrucosospora* wild type (WT), *A. verrucosospora* Δmad10, and *A. verrucosospora* Δmad10 pIB139-*mad10*. **c,** Extracted m/z = 783.8 HPLC-MS spectra from *A. verrucosospora* WT, *A. verrucosospora* Δmad31, and *A. verrucosospora* Δmad31 pIB139-*mad31*. All data are representative of three independent experiments.

Although *A. verrucosospora* Δmad10 no longer produced tetromadurin, analysis of the total ion current (TIC) and photodiode array (PDA) spectrum of this mutant revealed that it produced a new metabolite that we named T-17 (**Fig 8A and 8B**).

T-17 was produced by *A. verrucosospora* Δmad10 at ca. 50% the level of tetromadurin production by wild type *A. verrucosospora*. The mass spectra of T-17 contains peaks for putative

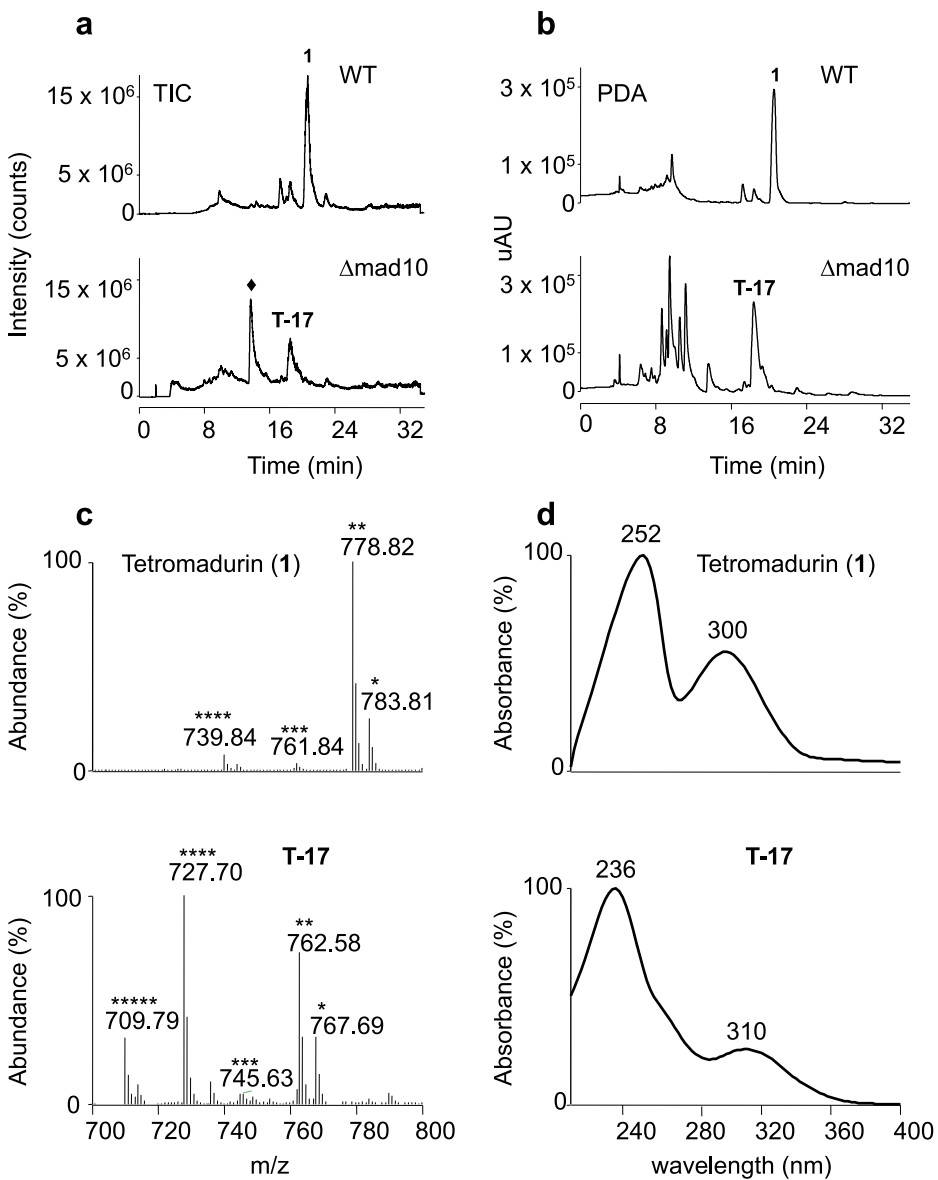

**Fig 8. Novel tetromadurin analogue produced by *A. verrucosospora* Δmad10. a,** Total ion current (TIC) spectra of the crude organic extracts from *A. verrucosospora* wild type (WT) and *A. verrucosospora* Δmad10. The other peaks represent unrelated species. Tetromadurin (**1**) was produced by *A. verrucosospora* and eluted at 19.8 min. A novel tetromadurin-related metabolite (named T-17) was produced by *A. verrucosospora* Δmad10, eluting at 17.4 min. **b,** HPLC-PDA (photodiode array) spectra of the crude organic extracts from *A. verrucosospora* WT and *A. verrucosospora* Δmad10. **c,** Positive ion mode mass spectrum of tetromadurin (**1**) and T-17: *[M+Na]⁺, **[M+NH4]⁺, ***[M+H]⁺, ****[M+H−H₂O]⁺, *****[M+H−2.H₂O]⁺. For tetromadurin (**1**): m/z $[C_{42}H_{64}O_{12}+Na]^+$ = 783.81; m/z $[C_{42}H_{64}O_{12}+NH_4]^+$ = 778.82; m/z $[C_{42}H_{64}O_{12}+H]^+$ = 761.84; m/z $[C_{42}H_{64}O_{12}+H−H_2O]^+$ = 739.84. For T-17: m/z $[C_{42}H_{64}O_{11}+Na]^+$ = 767.69; m/z $[C_{42}H_{64}O_{11}+NH_4]^+$ = 762.58; m/z $[C_{42}H_{64}O_{11}+H]^+$ = 745.63; m/z $[C_{42}H_{64}O_{11}+H−H_2O]^+$ = 727.70; m/z $[C_{42}H_{64}O_{11}+H−H_2O]^+$ = 727.70; m/z $[C_{42}H_{64}O_{11}+H−2.H_2O]^+$ = 709.79. **d,** UV absorption spectra of tetromadurin (**1**) ($\lambda_{max}$ = 252 nm, 300 nm, MeOH) and T-17 ($\lambda_{max}$ = 236 nm, 300 nm, MeOH). The ♦ indicates an unrelated compound that is also detectable in the WT strain.

[M+H]⁺ (m/z = 745.6), [M+NH₄]⁺ (m/z = 762.6), and [M+Na]⁺ (m/z = 767.7) ions, all of which are 16 Daltons less than the equivalent ions of tetromadurin (**Fig 8C and 8D**). A mass difference of 16 is diagnostic of an absent hydroxyl group, indicating that T-17 has the molecular formula $C_{42}H_{64}O_{11}$. Likely candidates for this missing hydroxyl are either the C36 or the

C38 primary hydroxyl group, predicted to be added by the cytochrome P450 enzymes Mad29 and Mad30, respectively. Closer examination of the mass/UV spectra of T-17 revealed striking similarities to the spectra of tetronasin intermediate **17** produced by *S. longisporoflavus* Δtsn11, in which the cyclohexane and tetrahydropyran rings are unformed [7]. Like **17**, the $UV_{max}$ of T-17 is also 236 nm (**Fig 8D**) [7]. Furthermore, also as reported for **17**, the major ion of T-17 detected is the $[M+H-H_2O]^+$ adduct (m/z = 727.7) [7]. In contrast, the major ion detected for tetronasin and tetromadurin in organic extracts of their respective producer strains is the $[M+Na]^+$ adduct (**Fig 8**). These data indicate that in addition to lacking one of the primary hydroxyl groups added by a cytochrome P450, T-17 also contains a labile hydroxyl group that is readily lost in the mass spectrometer. The likely explanation for this observation is that T-17 is the tetromadurin equivalent of tetronasin intermediate **17**, containing a labile hydroxyl group due to the yet-unformed tetrahydropyran and cyclohexane rings (**Fig 9**). It appears that in the absence of the cyclohexane and tetrahydropyran rings, one of the cytochrome P450-catalysed hydroxylations, either at C36 (**15**) or C38 (**16**), is unable to occur (**Fig 9A and 9B**).

In the tetronasin intermediate **17,** C30 bears a hydroxy group, indicating that cytochrome P450-catalysed hydroxylation of this carbon is not dependent on the cyclohexane and tetrahydropyran rings being present (**Fig 9C**). C36 is the equivalent position to C30 in **17**, offering a strong argument in favour of T-17 having the structure **15.** The second hydroxylation at C38

**Fig 9. Possible structures of T-17.** *A. verrucosospora* Δmad10 no longer produces tetromadurin but produces a tetromadurin derivative, T-17. The mass spectrum of T-17 indicates that it is missing a primary hydroxyl group. The primary hydroxyl groups of tetromadurin are located at C36 and C38. Two possible structures of T-17 are therefore possible where either: **a,** the C38 hydroxyl group (**15**) or **b,** the C36 hydroxyl group (**16**) is absent. **c,** the structure of the tetronasin derivative **17** produced by *Streptomyces longisporoflavus* Δtsn11 [7].

must occur after the cyclohexane ring has formed, suggesting it could be the final enzymatic step in tetromadurin biosynthesis (as has been depicted in Fig 5). Alternatively, the hydroxylation may precede tetrahydropyran formation.

Understanding the substrate tolerance of Mad10 and Mad31 could be important for using them as synthetic biology tools to create novel polyether tetronate antibiotics. In our previous study on tetronasin biosynthesis, we showed that Tmn9 and Tmn8 from the tetronomycin biosynthesis pathway could not substitute for Tsn11 and Tsn15, respectively [7]. Such a result is perhaps unsurprising, given that tetronasin and tetronomycin possess opposite configuration at their cyclohexane and tetrahydropyran rings. On the other hand, given that the stereochemistry of the cyclohexane and tetrahydropyran rings in tetromadurin is identical to tetronasin, we reasoned Mad10 and Mad31 would have a greater chance of successfully substituting for the equivalent *tsn* enzymes. To test this, *mad10* was expressed *in trans* in the *S. longisporoflavus* Δtsn11 mutant and *mad31* was expressed *in trans* in the *S. longisporoflavus* Δtsn15 mutant. However, HPLC analysis of these *S. longisporoflavus* mutants revealed that tetronasin production was not rescued in either case (S19 Fig), indicating that Mad10 and Mad31 are unable to accept the tetronasin intermediates.

## Discussion

In this work we have identified the biosynthetic gene cluster responsible for tetromadurin biosynthesis in *Actinomadura verrucosospora*. Our bioinformatics-based analysis of the *mad* gene cluster enabled almost of the genes to be assigned plausible roles in tetromadurin biosynthesis. Functional validation of the *mad* gene cluster was achieved by performing in-frame deletions in the putative cyclase genes *mad10* and *mad31*. The production of tetromadurin was abolished in both mutants, with the *A. verrucosospora* Δmad10 mutant producing a new compound, T-17, that mass spectrometry data supports as being an intermediate lacking the cyclohexane and tetrahydropyran rings, and one cytochrome P450-catalysed hydroxylation. The production of T-17 by the *A. verrucosospora* Δmad10 mutant provides experimental evidence that the enzymatic mechanism for cyclohexane and tetrahydropyran formation first described in tetronasin biosynthesis [7] is conserved in tetromadurin biosynthesis. The missing primary hydroxyl group of T-17 also sheds light on the timing of events in the biosynthesis pathway, indicating that this second P450-catalysed hydroxylation can only occur following the Mad10 reaction. While comparison with tetronasin intermediate **17,** isolated from *S. longisporoflavus* Δtsn11 provides convincing evidence that the structure of T-17 is **15**, we cannot rule out **16** being the true structure.

One feature of note in the polyketide synthase enzymes encoded in the *mad* gene cluster (MadAI-MadAVII) is the apparently inactive KR domain of module 12. This KR domain should reduce the C7 keto group to a D-hydroxyl group. One possible explanation could be that the KR domain of an adjacent module (either 11 or 13) performs this reduction. Domains acting externally to the PKS module they are located in have been proposed for other polyketide biosynthesis pathways [19, 64–66]. The KR domain of either module 11 or 13 could therefore be responsible for forming the C7 hydroxyl, though additional work is required to confirm this.

In addition to studying tetromadurin biosynthesis, a *S.* sp NRRL 11266 Δtmn8 mutant was created. Following the deletions of *mad10* and *mad31* in *Actinomadura verrucosospora*, *tmn8* was the only [4+2] cyclase homologue gene from one of the known polyether tetronate biosynthesis pathways that had not been deleted [7, 8]. The abolition of tetronomycin production in the *S.* sp NRRL 11266 Δtmn8 mutant confirms that both the VstJ-like and PyrE3-like cyclase homologues are essential for the biosynthesis of all three polyether tetronates, suggesting a

conserved mechanism of cyclohexane and tetrahydropyran formation. Homologues of these two cyclase classes will likely be encoded in the biosynthetic gene clusters of any related polyether tetronates yet to be discovered.

## Supporting information

**S1 Fig. Proposed biosynthetic precursors of the tetromadurin carbon backbone.** The carbon backbone of tetromadurin appears to be derived from six malonyl-CoA derived acetate units (red), eight (2S)-methylmalonyl-CoA derived propionate units (blue), one glyceryl-ACP (magenta), and one (2R)- methoxymalonyl-ACP (orange).
(EPS)

**S2 Fig. Production of tetromadurin by *Actinomadura verrucosospora*. a,** *A. verrucosospora* grew on oatmeal agar as pale colonies that formed white hydrophobic spores after several days. **b,** Structure of tetromadurin ($C_{42}H_{64}O_{12}$), m/z $[C_{42}H_{64}O_{12}+NH_4]^+$ = 778.8; m/z $[C_{42}H_{64}O_{12}+Na]^+$ = 783.8. **c,** Top: total ion current (TIC) chromatogram of *A. verrucosospora* organic extract. A peak corresponding to tetromadurin $[C_{42}H_{64}O_{12}+NH_4]^+$ (m/z = 778.8) was visible on the TIC at 19.8 min. Bottom: extracted m/z = 783.8 chromatogram of the tetromadurin sodium ion adduct $[C_{42}H_{64}O_{12}+Na]^+$ **c,** Mass spectrum of the tetromadurin peak from the TIC spectrum. *$[C_{42}H_{64}O_{12}+Na]^+$; **$[C_{42}H_{64}O_{12}+NH_4]^+$; ***$[C_{42}H_{64}O_{12}+H]^+$. **e,** UV absorption spectrum of the tetromadurin peak. $\lambda_{MAX}$ = 252 nm, 300 nm (MeOH), as previously reported [1].
(EPS)

**S3 Fig. Alignment of the KS domains encoded in the tetromadurin biosynthetic gene cluster.** The KS domain of the loading module (MadAI_AT_LM) and each of the 14 extension modules (KS_1–14) from the tetromadurin biosynthetic gene cluster were aligned using ClustalOmega[15]. The amino acids of the C-H-H catalytic triad are highlighted with an asterisk (*). The KS domain of the loading module contains a glutamine in place of the catalytic cysteine (highlighted in yellow).
(EPS)

**S4 Fig. Alignment of the AT domains encoded in the tetromadurin biosynthetic gene cluster. a,** The AT domain of the loading module (MadAI_AT_LM) and each of the 14 extension modules (AT_1–14) from the *mad* gene cluster were aligned using ClustalOmega [15]. The amino acid motif that predicts the selectivity of the AT domain is highlighted in either red (acetate-incorporating) or yellow (propionate-incorporating). **b,** The hypothetical linear tetromadurin intermediate predicted by retrobiosynthetic analysis. Propionate units derived from (2R)-methoxymalonyl-ACP are coloured in yellow. Acetate units derived from malonyl-CoA are coloured in red. The methoxy group predicted to arise from (2R)-methoxymalonyl-ACP is coloured in orange.
(EPS)

**S5 Fig. Alignment of the ER domains encoded in the tetromadurin biosynthetic gene cluster. a,** The ER domains from the tetromadurin biosynthetic gene cluster were aligned using ClustalOmega [15]. The tyrosine residue associated with the L-configuration at the α-position in the product is marked with an asterisk (*). The ER domains are highlighted either yellow (producing L-configuration at the α position) or red (producing D-configuration at the α position). Only MadAII_ER_4 and MadAII_ER_11 affect the final stereochemistry of tetromadurin, as their respective modules incorporate a propionate unit with an α methyl group. **b,** The linear tetromadurin intermediate predicted from its final structure. The fully saturated

carbon atoms (C9, C15, C23, and C27) produced by an ER domain are highlighted either in yellow (ʟ-configuration-determining ER) or red (ᴅ-configuration-determining ER).
(EPS)

**S6 Fig. Alignment of the DH domains encoded in the tetromadurin biosynthetic gene cluster. a,** The DH domains of the extension modules 1–13 from the *mad* gene cluster were aligned using ClustalOmega [15]. The tyrosine of the **Y**GP motif and the catalytic aspartic acid in the HPALL**D**AAL motif are marked with an asterisk (*). DH domains predicted to be active are highlighted in red and those predicted to be inactive are highlighted in yellow. **b,** The linear tetromadurin intermediate predicted from its final structure. The double bonds produced by DH domains are highlighted in red.
(EPS)

**S7 Fig. Alignment of the KR domains encoded in the tetromadurin biosynthetic gene cluster. a,** The KR domains of the extension modules 1–14 from the *mad* gene cluster were aligned using ClustalOmega [15]. The key features of a B1 type KR domain are highlighted: the (L/V/I) DD motif within the NADPH binding site (LDD), the catalytic tyrosine (*), and the locus of the proline found in A type KRs (P). All of the KR domains except KR_12 and KR_14 are predicted to be type B1. KR_12 and KR_14 both lacked the catalytic tyrosine residue, indicating they are inactive. KR_12 also had a significant deletion in its NADPH binding site. **b,** The linear tetromadurin intermediate predicted from its final structure. The three ᴅ-configured hydroxyls (C7, C17, and C21) predicted to form from a B1 type KR domain are highlighted in red. The ᴅ configuration of the methyl groups attached to C6 and C16 (C38 and C35, respectively) is also predicted to be governed by B1 type KRs. The KR domain in the position corresponding to the C7 hydroxyl (MadAVI_KR_12) appears to be inactive, so the origin of this hydroxyl is unknown.
(EPS)

**S8 Fig. Proposed biosynthesis of the tetromadurin tetronate ring.** The *mad* gene cluster contains the five genes of the glycerate-utilisation operon responsible for tetronate ring biosynthesis. The FkbH-like Mad7 catalyses likely formation of glyceryl-S-ACP using 1,3-bisphosphoglycerate and the ACP, Mad8. The FabH-like protein Mad16 then catalyses C-C and C-O bond formation to form the tetronate ring and release the polyketide chain from the PKS. Finally, the C40-C41 exocyclic double bond is formed *via* acetylation of the C41 hydroxyl by acetyltransferase Mad17, followed by elimination catalysed by the dehydratase Mad18.
R = linear tetromadurin intermediate.
(EPS)

**S9 Fig. Proposed biosynthesis of (2*R*)-methoxymalonyl-ACP.** The FkbH-like Mad14 likely catalyses formation of glyceryl-ACP from 1,3 bisphosphoglycerate and the acyl-carrier protein Mad12. Glyceryl-ACP then undergoes two oxidation steps, catalysed by the dehydrogenases Mad11 and Mad13 to form (2*R*)-hydroxymalonyl-ACP, which is methylated by Mad15.
(EPS)

**S10 Fig. Proposed biosynthesis of the tetrahydrofuran rings in tetromadurin.** MadC catalyses the regio- and stereospecific epoxidation of the C24-C25 and C28-C29 *E* double bonds. Next, the MadB epoxide hydrolase catalyses two sequential *exo* ring-closures to form the two tetrahydrofuran rings.
(EPS)

**S11 Fig. Proposed biosynthesis of the cyclohexane and tetrahydropyran rings in tetromadurin.** Mad10 catalyses an inverse-electron demand hetero-Diels-Alder reaction to form

oxadecalin-contain intermediate **10** from **9**. The equivalent oxadecalin from the tetronasin biosynthesis pathway, **17**, is hydrated to form a hemiacetal [7], so the tetromadurin oxadecalin intermediate may do the same (forming **11**) given its structural similarity. Mad31 then catalyses a pericyclic rearrangement to form the tetrahydropyran ring and dismantle the cyclohexane ring, using the non-hemiacetal form of the oxadecalin intermediate **10** (as also predicted for tetronasin biosynthesis [7]).
(EPS)

**S12 Fig. Alignment of PyrE3-type [4+2] cyclases.** Protein sequence alignment of Mad10 (QKG20158 *see note in methods section relating to the revised sequence), Tmn9 (BAE93724), and Tsn11 (ACR50781) [7], the [4+2] cyclase TedJ (KYG52530) from the tetrodecamycin BGC [67], The [4+2] cyclase KijA (ACB46484) from the kijanimicin BGC [45]; The [4+2] cyclase PyrE3 (AFV71312) from the pyrroindomycin BGC [42]; The [4+2] cyclase VstK (BAQ21949) from the versipelostatin BGC [46], The [4+2] cyclase ChlE3 (AAZ77700) from the chlorothricin BGC [48]; The monooxygenase PgaE (AAK57522) from gaudimycin C biosynthesis [50]; The monooxygenase OxyS (4K2X_A) from oxytetracycline biosynthesis [49]; the monooxygenase MtmOIV (CAK50794) from mithramycin biosynthesis [43]. Red stars (★) indicate amino acids involved in binding FAD [49]. Black asterisks (*) indicate the locations of the four arginine residues in MtmOIV involved in NADPH binding [43]. Similar residues in an alignment are coloured accordingly: Blue, hydrophobic; Green, polar; Purple, negative; Red, positive; Yellow, proline; Orange, glycine. Alignment was made using ClustalOmega [15].
(EPS)

**S13 Fig. Maximum likelihood phylogenetic tree of PyrE3 homologues.** A maximum likelihood phylogenetic tree was created for the diverse family of PyrE3 homologues, including: [4+2] cyclases from spirotetronate/tetramate pathways; FAD-dependent monooxygenases; and homologues from polyether tetronate pathways. 1000 bootstrap replicates were performed.
(EPS)

**S14 Fig. Alignment of VstJ-type [4+2] cyclases.** Protein sequence alignment of [4+2] cyclases encoded in the biosynthetic gene clusters of the polyether tetronates tetronasin (Tsn15: 6NOI_A), tetronomycin (Tmn8: BAF73716), and tetromadurin (Mad31: QKG20137); the spirotetronates abyssomicin (AbyU: 5DYV_A), versipelostatin (VstJ: BAQ21945), chlorothricin (ChlL: AAZ77701) and the spirotetramate pyrroindomycin (PyrI4: AFV71338). Similar residues in an alignment are coloured accordingly: Blue, hydrophobic; green, polar; purple, negative; red, positive; yellow, proline; orange, glycine. Alignment was made using ClustalOmega [15]. The (*) marks the conserved tryptophan.
(EPS)

**S15 Fig. Proposed cytochrome P450-catalysed hydroxylations in tetromadurin biosynthesis.** Mad29 is a homologue of the cytochrome P450 enzymes Tsn12 and Tmn14 present in the tetronasin and tetronomycin BGCs, respectively. We therefore propose that that Mad29 catalyses hydroxylation of C36, leaving Mad30 to hydroxylate C38.
(EPS)

**S16 Fig. Creation of the *A. verrucosospora* Δmad10 and *A. verrucosospora* Δmad31 deletion mutants.** a, Creation of the *A. verrucosospora* Δmad10. Left: Diagram showing the double crossing-over event between pYH7-*mad10* and the *A. verrucosospora* chromosome. In total 906/1404 bp of *mad10* were deleted in the Δmad10 mutant. The primers mad10_KO_Fw and mad10_KO_Rv were used to screen the genomic DNA of double crossover exconjugants to

identify those containing the *mad10* coding-frame deletion. PCR product size wild type: 1913 bp. PCR product size Δmad10 mutant: 1007 bp. Right: Agarose gel demonstrating the successful creation of the *A. verrucosospora* Δmad10 mutant. b, Creation of the A. verrucosospora Δmad31 mutant. Left: Diagram showing the double crossing-over event between pYH7-*mad31* and the *A. verrucosospora* chromosome. In total 291/561 bp of *mad31* were deleted in the Δmad31 mutant. The primers mad31_KO_Fw and mad31_KO_Rv were used to screen the genomic DNA of double crossover exconjugants to identify those containing the Δmad31 coding-frame deletion. PCR product size wild type: 980 bp. PCR product size Δtsn15 mutant: 689 bp. Right: Agarose gel demonstrating the successful creation of a *S. longisporoflavus* Δmad31 mutant.

(EPS)

**S17 Fig. Creation of the *S*. sp NRRL 11266 Δtmn8 deletion mutant.** Left: Diagram showing the double crossing-over event between pYH7-*tmn8* and the *S*. sp NRRL 11266 chromosome. In total 573/573 bp of *tmn8* were deleted in the Δtmn8 mutant. The primers tmn8_KO_Fw and tmn8_KO_Rv were used to screen the genomic DNA of exconjugants to identify those in which the coding-frame deletion had taken place. PCR product size wild type: 971 bp. PCR product size Δtmn8 mutant: 398 bp. Left: Agarose gel demonstrating the successful creation of the *S*. sp NRRL 11266 Δtmn8 mutant.

(EPS)

**S18 Fig. HPLC-MS analysis of the *Streptomyces* sp. NRRL 11266 Δtmn8 mutant. a,** The structure and molecular weight of tetronomycin (**3**) produced by *S*. sp. NRRL 11266. **b,** Extracted m/z = 609 chromatograms of *S*. sp. NRRL 11266 wild type (WT) and *S*. sp. NRRL 11266 Δtmn8. Tetronomycin **3** was detected in *S*. sp. NRRL 11266 WT as ammonium (m/z = 604.42) and sodium (m/z = 609.5) adducts. No tetronomycin production was detected from the *S*. sp. NRRL 11266 Δtmn8 mutant. Data are representative of three independent experiments.

(EPS)

**S19 Fig. *in vivo* cross-complementation using *tsn11* and *tsn15* homologues. a,** Genetic complementation of the *S. longisporoflavus* Δtsn11 mutant. I) Production of tetronasin **2** by *S. longisporoflavus* WT. II) Production of **17** by *S. longisporoflavus* Δtsn11. III) Production of **17** by *S. longisporoflavus* Δtsn11 pIB139-*mad10*. IV) Production of tetronasin **2** by *S. longisporoflavus* Δtsn11 pIB139-*tsn11*. Data are representative of three independent experiments. **b,** Genetic complementation of the *S. longisporoflavus* Δtsn15 mutant. I) Production of tetronasin **2** by *S. longisporoflavus* WT. II) Lack of tetronasin production by the *S. longisporoflavus* Δtsn15 mutant. III) Lack of tetronasin production by the *S. longisporoflavus* Δtsn15 pIB139-*mad31* mutant. IV) Production of tetronasin **2** by *S. longisporoflavus* Δtsn15 pIB139-*tsn15*. Data representative of three independent experiments.

(EPS)

## Acknowledgments

The authors thank the DNA sequencing facility of the Department of Biochemistry, University of Cambridge, UK.

## Author Contributions

**Conceptualization:** Rory F. Little, Peter F. Leadlay.

**Data curation:** Markiyan Samborskyy.

**Formal analysis:** Rory F. Little, Markiyan Samborskyy.

**Funding acquisition:** Rory F. Little, Peter F. Leadlay.

**Investigation:** Rory F. Little.

**Methodology:** Rory F. Little, Peter F. Leadlay.

**Project administration:** Peter F. Leadlay.

**Resources:** Peter F. Leadlay.

**Software:** Markiyan Samborskyy.

**Supervision:** Peter F. Leadlay.

**Validation:** Rory F. Little, Peter F. Leadlay.

**Writing – original draft:** Rory F. Little.

**Writing – review & editing:** Markiyan Samborskyy, Peter F. Leadlay.

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
