## [Decision Letter · Decision Letter 0]

20 May 2020

PONE-D-20-10863

The biosynthetic pathway to tetromadurin (SF2487/A80577), a polyether tetronate antibiotic.

PLOS ONE

Dear Dr Little,

Thank you for submitting your manuscript to PLOS ONE. After careful consideration, we feel that it has merit but does not fully meet PLOS ONE’s publication criteria as it currently stands. Therefore, we invite you to submit a revised version of the manuscript that addresses the points raised during the review process.

We would appreciate receiving your revised manuscript by Jul 04 2020 11:59PM. To enhance the reproducibility of your results, we recommend that if applicable you deposit your laboratory protocols in protocols.io, where a protocol can be assigned its own identifier (DOI) such that it can be cited independently in the future. For instructions see: http://journals.plos.org/plosone/s/submission-guidelines#loc-laboratory-protocols

We look forward to receiving your revised manuscript.

Kind regards,

Christopher N. Boddy, Ph.D.

Academic Editor

PLOS ONE

Journal Requirements:

2. We note you have included tables to which you do not refer in the text of your manuscript. Please ensure that you refer to Tables 1, 2 and 3 in your text; if accepted, production will need this reference to link the reader to the Tables.

Additional Editor Comments (if provided):

Reviewers' comments:

Reviewer's Responses to Questions

**Comments to the Author**

1. Is the manuscript technically sound, and do the data support the conclusions?

Reviewer #1: Yes

Reviewer #2: Yes

2. Has the statistical analysis been performed appropriately and rigorously? 

Reviewer #1: Yes

Reviewer #2: Yes

3. Have the authors made all data underlying the findings in their manuscript fully available?

Reviewer #1: Yes

Reviewer #2: Yes

4. Is the manuscript presented in an intelligible fashion and written in standard English?

Reviewer #1: Yes

Reviewer #2: Yes

5. Review Comments to the Author

Reviewer #1: This manuscript describes the identification and characterization of the biosynthetic gene cluster for tetromadurin from Actinomadura verrucosispora. The authors present a thorough biosynthetic proposal based upon bioinformatic analysis of the gene cluster alongside several strategic gene inactivation experiments. Comparisons are drawn to the previously studied tetronasin and tetronomycin pathways.

The manuscript is well written with only a couple of very minor typographic errors and describes sound, high quality experimental research.

Publication is recommended after these minor changes.

Line 81-83: How were the necessary building blocks determined? Was this previously reported? Based on labelling experiments? Or is this a prediction?

Line 192: Add information about the MS parameters used for HPLC-MS analysis

Line 213-214: Include information in methods section about how the Actinomadura verrucosispora was sequenced. Include gDNA isolation and sequencing preparation steps as well as genome assembly information. If this was carried out by a service, state which company/lab

Figures 4/5: would benefit from using colour to highlight chemical changes in the structures. Also hard to read the writing inside circles in Fig 4, remove shading in circles.

Line 282-285: These two sentences feel out of place a confusing. Either remove entirely or rephrase and move to after the description of the PKS genes that are present.

Line 300-303: The use of the term “predicted” to describe both of the structures in Fig 6 caption is confusing and inconsistent with the statement in line 292-294 in the text. Please clarify language

Line 421: Change “(Fig 8c)” to “(Fig 9c)”

Line 500-502: Can you really use this argument if the complementation worked for Mad10 using the same vector?

Line 533: Change “Fig 7” to “Fig 8”

Line 756: Change “oolyether” to “polyether”

Reviewer #2: Little et al identified the biosynthetic gene cluster involved in polyether compound tetromadurin biosynthesis in Actinomadura verrucosispora. They did bioinformatic analysis and performed gene deletions (cyclase genes mad10 and mad31) to confirm the involvement of this cluster in tetromadurin biosynthesis. The overall writing is clear. The following issues are suggested to be addressed:

The biggest issue is that the novel tetromadurin analogue produced by Δmad10 was not purified and identified by NMR analysis.

Fig 4, 5: please combine these two figures. For Fig 4, I would like to suggest the authors to use different colors to indicate the specificities of the ATs.

Fig 6: It is more appropriate to remove Fig 6 to supporting information as this product is predicted via bioinformatics analysis.

In the section of “Genes for (2R)-methoxymalonyl-ACP biosynthesis”, line 381-382, tautomycin and oxazolomycin are suggested to be referred as well.

Fig 9: It is more appropriate to remove Fig 9 to supporting information.

S13 Fig: more PyrE3 homologues are suggested to be subjected to phylogenetic analysis.

Line 987: “Fig Alignment of” should be “S14 Fig Alignment of” The accession numbers of protein sequences for alignment are suggested to be added.

Please number all the compounds in the manuscript.

6. PLOS authors have the option to publish the peer review history of their article (what does this mean?). If published, this will include your full peer review and any attached files.

Reviewer #1: No

Reviewer #2: No

---

## [Author Response · Author response to Decision Letter 0]

3 Jul 2020

Please see attached .pdf document for our response.

---

## [Editor Report · Decision Letter 1]

31 Aug 2020

The biosynthetic pathway to tetromadurin (SF2487/A80577), a polyether tetronate antibiotic.

PONE-D-20-10863R1

Dear Dr. Little,

We’re pleased to inform you that your manuscript has been judged scientifically suitable for publication and will be formally accepted for publication once it meets all outstanding technical requirements.

Kind regards,

Patrick C. Cirino

Academic Editor

PLOS ONE
---

## [Editor Report · Acceptance letter]

3 Sep 2020

PONE-D-20-10863R1 

The biosynthetic pathway to tetromadurin (SF2487/A80577),
a polyether tetronate antibiotic 

Dear Dr. Little:

I'm pleased to inform you that your manuscript has been deemed suitable for publication in PLOS ONE. Congratulations! Your manuscript is now with our production department. 

Kind regards, 

on behalf of

Dr. Patrick C. Cirino 

Academic Editor

PLOS ONE